# Therapeutic Effects of Cannabinoids and Their Applications in COVID-19 Treatment

**DOI:** 10.3390/life12122117

**Published:** 2022-12-15

**Authors:** Rebeca Pérez, Talita Glaser, Cecilia Villegas, Viviana Burgos, Henning Ulrich, Cristian Paz

**Affiliations:** 1Laboratory of Natural Products & Drug Discovery, Center CEBIM, Department of Basic Sciences, Universidad de La Frontera, Temuco 4811230, Chile; 2Department of Biochemistry, Instituto de Química, Universidade de São Paulo, Av. Prof. Lineu Prestes 748, São Paulo 05508-000, SP, Brazil; 3Departamento de Ciencias Básicas, Facultad de Ciencias, Universidad Santo Tomas, Temuco 4780000, Chile

**Keywords:** refractory epilepsy, COVID-19, phytocannabinoids, synthetic cannabinoids, clinical trials

## Abstract

*Cannabis sativa* is one of the first medicinal plants used by humans. Its medical use remains controversial because it is a psychotropic drug whose use has been banned. Recently, however, some countries have approved its use, including for recreational and medical purposes, and have allowed the scientific study of its compounds. *Cannabis* is characterized by the production of special types of natural products called phytocannabinoids that are synthesized exclusively by this genus. Phytocannabinoids and endocannabinoids are chemically different, but both pharmacologically modulate CB1, CB2, GRP55, GRP119 and TRPV1 receptor activities, involving activities such as memory, sleep, mood, appetite and motor regulation, pain sensation, neuroinflammation, neurogenesis and apoptosis. Δ9-tetrahydrocannabinol (THC) and cannabidiol (CBD) are phytocannabinoids with greater pharmacological potential, including anti-inflammatory, neuroprotective and anticonvulsant activities. Cannabidiol is showing promising results for the treatment of COVID-19, due to its capability of acting on the unleashed cytokine storm, on the proteins necessary for both virus entry and replication and on the neurological consequences of patients who have been infected by the virus. Here, we summarize the latest knowledge regarding the advantages of using cannabinoids in the treatment of COVID-19.

## 1. Introduction 

The use of cannabis products has dramatically increased in recent times, due to the lack of response to standard treatments, for example to anxiety, chronical inflammation problems and seizures, which puts patients’ lives at risk. The hemp plant has been used for centuries in medicinal and recreational products such as hashish or marijuana. Cannabis use for medicinal purposes is legal in some countries, including Germany, Austria, Canada, Spain, Finland, Israel, Italy, Holland, Portugal and some states of the United States of America [1]. In Latin American countries, its use is permitted in Uruguay, Colombia and Argentina [2]. In Chile, the sale of cannabis-based medicines was authorized after the modification of a restricting law in 2015. For these reasons, it is expected that by 2025 the global medical cannabis market will reach $43 billion [3]. The positive effects of the medicinal use of marijuana in conditions such as chronic pain, multiple sclerosis, nausea, vomiting, terminal diseases and epilepsy have been reported [1,2]. Several studies suggest that cannabidiol (CBD) may be useful in the treatment of patients with COVID-19 by inhibiting the expression of the protein critical for SARS-CoV2 virus replication and by binding as an agonist to the cannabinoid receptor 2 (CB-2), thus reducing inflammation and lung damage [4,5,6,7].

To date, seven strains of human-infecting coronaviruses have been identified [8]. The currently available treatment is supportive, as vaccination prevents the severe illness and death of patients, but its efficacy in controlling virus spread and preventing future deaths is not clear yet [9]. For these reasons, it has become a global public health emergency, making it extremely important to discover new alternatives for effective treatment. 

Despite the few studies that exist on the antiviral activity of cannabinoids, some research has been conducted that demonstrates their high anti-inflammatory and immunosuppressive power [2,10]. Thus, by inhibiting both the coronavirus replication processes and the inflammation provoked, there is the possibility of developing an effective therapeutic strategy [6,7]. The aim of this article is to review the therapeutic applications of cannabis derivatives and their applications for the treatment of COVID-19, based on the mechanisms through which they exert their pharmacological activity.

## 2. Endocannabinoids

The endocannabinoid system is a neuromodulatory network involved in many cognitive and physiological processes, such as in anti-convulsant, anti-inflammatory, anti-depressant and anti-tumorigenic actions, as well as exerting analgesic, hyperphagic, hypophagic, neuroprotective and antiemetic activities [11] and appetite regulation [12]. The endocannabinoid system is composed of endogenous cannabinoids (chemical derivatives of arachidonic acid), cannabinoid receptors and enzymes that produce and degrade endocannabinoids [13,14].

These natural molecules are derived from membrane phospholipids, which include arachidonic acid as endogenous cannabinoids present in the organism of all animals. They are synthesized in neurons on demand and released into the synaptic space, where they activate membrane receptors, acting in the vicinity of the site where they are released, regulating synaptic transmission. The principal endocannabinoids are anandamide (AEA) and 2-arachidonoyl glycerol (2-AG), but they also include 2-arachidonoyl-glycerol (2-AGE), O-arachidonoyl-ethanolamine (virodamine) and N-arachidonoyl-dopamine (NADA) [11]. The structures of endocannabinoids are given in Figure 1.

Anandamide (AEA) was the first discovered endocannabinoid from pig brain in 1992 [15]. Anandamide is produced from *N*-arachidonoyl phosphatidylethanolamine, as shown in Figure 2. It behaves as a full or partial agonist of the cannabinoid receptor 1 (CB1) as well as a partial CB2 agonist. It is present in the CNS, spleen, heart, testis, uterus and vascular endothelium of humans [16]. 

In patients with autism spectrum disorder, the plasma concentrations of AEA are significantly lower compared to control children, suggesting that AEA signaling participates in the pathophysiology of autism, evaluated in 116 children around 3 to 12 years old [17]. The pharmacological increase in AEA levels results from the inhibition of fatty acid amide hydrolase (FAAH), which is responsible for AEA degradation to arachidonic acid (AA) and ethanolamine. Elevated levels of AEA may aid the treatment of post-traumatic stress disorder [18]. For example, in a double-blind, placebo-controlled study, the inhibition of FAAH (PF-04457845, 4 mg/once daily, over 10 days, *n* = 16) increased 10-fold in baseline anandamide concentration, with an improvement of fear extinction memories in the patients and attenuation of the anxiogenic effects of stress [19]. Further, increased AEA concentrations caused by the inhibition of FAAH could promote learning during psychotherapeutic interventions [20]. On the other hand, AEA production appears to be deficient in patients with hypertrophic scars suggesting a correlation between the systemic and local skin endocannabinoid systems during human wound healing [21].

2-Arachidonoyl glycerol (2-AG) was the second endocannabinoid to be discovered. It is the most abundant endogenous cannabinoid in the CNS. It is an ester derived from arachidonic acid and glycerol, produced by the enzyme diacylglycerol lipase alpha and beta (DAGL). The inhibition of the actions of this endocannabinoid is mediated by hydrolysis with the enzyme monoacylglycerol lipase (MAGL) producing arachidonic acid (AA) and glycerol (Figure 3). 2-AG behaves as a full agonist of the CB1 and CB2 receptors. However, it also binds to other receptors with cannabimimetic cellular activity, such as GRP55 and TRPV1 receptors [16,22]. 

Alternatively to the aforementioned hydrolysis routes, AEA and 2AG can be oxidized by cyclooxygenase-2 (COX-2), different lipoxygenases or cytochrome P450 [23]. 2-AG hydrolysis inhibition could be an anti-inflammatory strategy to increase 2-AG levels and decrease eicosanoids levels. Human leukocytes use AA and unsaturated fatty acids to biosynthesize 2-AG and other monoacylglycerols (MAGs), which might decrease inflammation processes [24].

## 3. Cannabinoid Receptors

In 1990, the first cannabinoid-specific receptor (CB1) was identified and cloned, followed by a second receptor (CB2). Cannabinoid receptor 1 and CB2 differ in their signal transduction and in their distribution in different tissues. The identification of these receptors led to a better understanding of phytocannabinoid and endocannabinoid functions and to the creation of synthetic cannabinoids [25]. Cannabinoid binding to their receptors results in distinct effects depending on which signaling pathway is activated. Signaling pathway activation depends on the agonist structure and the monomeric or heteromeric cannabinoid receptor states [26]. Furthermore, CB1 and CB2 receptors differentially regulate TNF-α-induced apoptosis in HT22 hippocampal cells, CB1 being a promoter of signaling and neuroprotection [27]. Once endocannabinoids have been released by the presynaptic neuron, the cannabinoid receptors are activated, inhibiting the release of neurotransmitters, such as glutamate [28]. This action of cannabinoids confers to them important physiological roles as regulators of synaptic activity. 

CB1 receptors are principally expressed in the neurons of the cortex, spinal cord and peripheral nervous system as well as in peripheral organs and tissues. These receptors are expressed in brain areas responsible for memory processing, movement, and pain modulation. Their activation results in different effects on pain sensation as well as on sleep, memory, appetite, mood, and motor regulation [29,30].

CB2 receptors are expressed by immune cells, such as leukocytes, in the spleen, tonsils and in the peripheral nervous system. The functions of CB2 receptors include the modulation of the immune system and inflammation by the release of cytokines. Since selective CB2 receptor agonists do not cause psychotropic effects, these are increasingly investigated for the therapeutic applications of cannabinoids, such as analgesics, anti-inflammatories and anti-neoplastics [29,30]. 

The GRP55 receptor is a putative cannabinoid receptor 3 (CB3). Experimental evidence links the GPR55 receptor to pathological conditions, providing a therapeutic target in cancer, neurodegenerative and metabolic disorders as well as in the regulation of microglia-mediated neuroinflammation [31]. The receptor has been found in the adrenal glands, spleen, intestine and CNS. The expression of the active receptor has been detected in bone cells [32,33]. GRP55 receptor activation occurs through phospholipase C (PLC)-β pathways. Consequently, there is an increase in cytosolic calcium concentration with the further stimulation of GTPases (Rac1, RhoA and Cdc42) and mitogen- or stress-activated protein kinases (MAPKs, ERK and p38) and NFAT, NFKB, CREB and ATF2 transcription factor activation [34,35,36]. GRP119 cannabinoid receptors are localized in humans predominantly in the pancreatic tissue and in the enteroendocrine L cells of the gastrointestinal tract. Their activation elevates plasma insulin concentrations in mice and decreases food intake and weight in rats. It has been proposed as the endogenous receptor for oleoyl ethanolamide [37]. 

TRPV1 cannabinoid receptors are vanilloid receptors and their activation promotes programmed death (apoptosis) [38] by augmenting intracellular calcium concentration, ceasing mitochondrial membrane potential, cytochrome C release and caspase 3 and 9 activation [39]. 

## 4. Phytocannabinoids

The genus *Cannabis* comprises the plants *Cannabis sativa*, *Cannabis indica*, *Cannabis americana* and *Cannabis ruderalis*. *C. sativa* grows in tropical and subtropical regions as well as temperate climate zones, e.g., in Europe [1,2]. The original plant has been known for 10,000 years in practically all cultures since the discovery of agriculture. Countries such as China, India, Turkestan and Arabia have used its compounds for medicinal purposes to treat diseases such as malaria, beriberi, constipation, rheumatic pains, headaches, female ailments and other conditions [11]. In the 19th century, its widespread use as a stimulant, sedative and analgesic became popular [30]. In 1860, the first committee of physicians in the USA was created to systematically study its use and properties. In 1964, delta-9 tetrahydrocannabinol (Δ9-THC), the main substance responsible for the psychoactive and pharmacological properties of marijuana extracts, was isolated. 

Cannabis plants contain over 500 bioactive compounds in their 18 different species, including over 100 different naturally occurring phytocannabinoids. Phytocannabinoids are produced in glandular hairs or trichomes. While they are found on most aerial surfaces of the plant, phytocannabinoids are primarily located in the bracts and flowers of the plant. [30] Phytocannabinoids are compounds of terpene phenolic structure with 21 carbon atoms. The main phytocannabinoids are Δ9-tetrahydrocannabinol (THC), Δ8-tetrahydrocannabinol (Δ8-THC), cannabidiol (CBD), cannabinol (CBN), and CBD being the known active compounds of pharmacological plant activity (Figure 4). 

Despite THC displaying the highest psychoactive potential, its pharmacological properties have been proven in clinical and preclinical assays as appetite stimulation, the suppression of chemotherapy-induced nausea and vomiting and the inhibition of pain, spasticity, neuroprotective, anti-inflammatory and antitumor effects [40,41,42].

Δ8-THC and CBD are found in some plant varieties and always in low concentrations. CBD also exerts psychoactive properties, and its action on the CB2 receptor in splenocytes and thymocytes causes a decrease in the transcription of the gene for interleukin-2 (IL-2) [43]. CBD has shown anxiolytic, anticonvulsant and antipsychotic effects. Further antioxidant, anti-inflammatory, antiemetic, antitumor and neuroprotective properties of CBD have been reported in preclinical studies. Cannabinol has analgesic properties with fewer psychoactive effects than Δ9-TCH. The activity of commercial formulations based on cannabinoids is summarized in Table 1.

## 5. Clinical Evidence for Therapeutic Efficacies of Cannabinoids

Cannabinoids are mostly used as anxiolytic, relaxing and anti-inflammatory natural agents with adjuvant properties for the treatment of epilepsy, schizophrenia, multiple sclerosis, depression or chronic pain [38,46,50,51,52,53,54,55]. In the case of epilepsy, it has been shown that the brain tissue of these patients shows an overexpression of proinflammatory cytokine IL-1β and IL-6 genes together with nuclear transcription factor kappa B (NFKB) [56]. Cannabinol can inhibit the G protein-coupled orphan receptor (GRP55) and decrease NFKB signaling, the latter probably by binding to nuclear PPAR-g receptors, thus reducing the expression of proinflammatory enzymes such as iNOS (nitric oxide synthase) and COX-2 (cyclooxygenase type 2) and metalloprotease and proinflammatory cytokine production [10,43,53,57,58]. 

There are numerous clinical trials proving the pharmacological efficacy of CBD, which are summarized in Table 2.

Another issue of great importance is the recently discovered interaction between first generation antiepileptic drugs (carbamazepine, oxcarbazepine, phenytoin, phenobarbital and primidone) and the decrease in the concentration of some drugs used for COVID-19 treatment (atazanavir and remdesivir, darunavir/cobicistat and lopinavir/ritonavir). Therefore, in these cases, the use of CBD to treat both conditions can be an alternative that can change the results obtained [64].

## 6. Cannabis and COVID-19: In Vitro Studies

As of 24 October 2022, there have been 624,235,272 confirmed cases of COVID-19, including 6,555,270 deaths, reported to the WHO, and a total of 12,814,704,622 doses of vaccine have been applied [65].

The seriousness of SARS-CoV2-caused illness lies in its capability of inducing in the infected person a rapid and intense immune response that courses in an uncontrolled manner, unleashing a storm of cytokines resulting in a variety of symptoms, including severe morbidity and mortality. This cytokine storm is produced by the organism attempting to defend itself against viral invasion, thus stimulating the production of macrophages, interleukins, chemokines and proinflammatory mediators [66]. The endocannabinoid system has the role of acting as a key regulator of the immune system, as it possesses the capacity of immunosuppression and repressing cytokine cascade, inhibiting immune cell proliferation and antibody production [7,67,68,69,70]. 

The angiotensin-converting enzyme 2 (ACE 2) has been pointed out as the principal receptor for the interaction of SARS-CoV 2 with human cells, which is mainly expressed in type I and II pneumocytes, making these cells are more susceptible of being infected by the virus. ACE 2 receptors are also expressed by lung AT2, liver cholangiocytes, colon colonocytes, esophageal keratinocytes, ileum EC, rectum EC, stomach epithelial cells and kidney proximal tubules [70]. CD169^+^ macrophages residing in the spleen and lymph node tissues can be infected [71,72,73,74,75]. Alveolar macrophages do not only protect the lungs from pathogens, but also participate in enhancing specific T-cell responses, repairing damaged tissue and recruiting neutrophils leading to increased inflammatory cell traction [76,77,78]. The hyperactivation of lung macrophages and proinflammatory monocyte-derived macrophages (MDM) occurs in the small airways. The transcriptomic analysis of plasma and bronchoalveolar lavage fluids from SARS-CoV2 patients has demonstrated large amounts of proinflammatory cytokines, chemokines and soluble inflammatory mediators, including TNF-α, IL-6, IL-1β, IL-2R, IL-8, inducible protein (IP)-10, C-reactive protein and D-dimer, culminating in a cytokine storm [79]. In the inflammation process, the overexpression of genes that express chemokines was observed, which are critical for recruiting neutrophils (CXCL17) and monocytes (CCL2, CCL7) in the lungs [80]. In SARS-CoV2 and MERS-CoV, a rapid and specific memory CD8 T-cell response is required to protect against infection [24,25]. However, severe SARS-CoV-2 infection results in CD4 and CD8 T-cell lymphopenia and a decrease in INF-γ-producing T cells [81,82].

## 7. Indispensable Proteins for the Development of Infection and CBD

The structural characteristics of SARS-CoV2 indicate that there are host cell proteins that are indispensable for the progress of viral infection. Once inside the lung cells, it binds to the host cell membrane receptor: angiotensin-converting enzyme 2 (ACE 2), via the transmembrane spike glycoprotein (structural glycoprotein of the virus) and induces its entry through membrane endocytosis. This glycoprotein forms homotrimers that protrude from the surface of SARS-CoV2. The subunits of the SARS-CoV2 spike glycoprotein trimer consist of an S1 subunit that binds to host cell ACE2 to initiate infection, an S2 subunit that mediates the fusion of the virus with host cells and a transmembrane domain [83]. While ACE2 is the receptor for viral entry, the transmembrane enzyme transmembrane serine protease 2 (TMPRSS2) primes viral spike proteins for SARS-CoV2 entry into host cells. Its function is given by proteolysis and the consequent activation of the spike protein by cleaving it into two covalently linked peptides: the S1 (ACE2 receptor-binding terminal) and S2 (viral cell membrane fusion-mediated terminal) sites or subunits, leading to conformational changes for the fusion of the virus with the host membrane and the virus entering of the cytoplasm [84,85,86]. The S1 subunit is primarily responsible for host virus range determination and cell tropism [71,87].

Then, ORF1, one of the nonstructural viral proteins, stimulates virus replication and RNA synthesis, while the nucleocapsid phosphoprotein (structural viral protein) packages the viral genome, forming a helical nucleocapsid with an RNA chaperone function. The surface glycoproteins evoke virion assembly and morphogenesis, producing virus particles and releasing them by exocytosis [66,88,89]. Any interference with the expression of one of these proteins may disrupt the transmission cycle of SARS-CoV2 [6]. 

Erukainure et al. demonstrated that *C. sativa* compounds have strong binding affinities to the initiation and termination codons of ORF1, the surface glycoprotein, the envelope protein and nucleocapsid phosphoprotein mRNAs of the complete SARS-CoV2 genome isolated from KwaZulu-Natal, South Africa [90]. This highlights the utility of phytocannabinoids in slowing down virus replication, translation, assembly, and release. The nature of these binding affinities could be due to their chemical properties. The cannabinoid compounds present in their structure free hydroxyl group and aromatic rings that enable them to interact with purines, pyrimidines or the phosphate terminus of the mRNA sequence of the SARS-CoV2 genome through hydrogen bonds or intermolecular interaction.

Purines and pyrimidines, as part of nucleotides, are capable of undergoing nucleophilic or electrophilic reactions, specifically the formation of hydrogen bonds between the oxygen at carbon 2 of the purines in the SARS-CoV2 mRNA genome [91]. The series of electrostatic interactions between the pyrimidines and the natural product could be due to the free amine bound to carbons 2 and 6 of guanine and adenine, respectively. Similarly, the polar region of phytocannabinoids could electrostatically interact with the phosphate terminal of SARS-CoV2 mRNA [92,93]. CBD inhibits SARS-CoV2 replication with an IC_50_ of 8 µM. In view of that, CBD is at least as potent as the antiviral drugs Remdesivir (RDV) and Lopinavir, which are employed for COVID-19 treatment [94,95,96,97].

CBD effectively eradicates viral mRNA expression in host cells. SARS-CoV2-induced gene expression reduction connected to chromatin modification and transcription was reversed. Gene expression reduction of mRNA spike, membrane, envelope and nucleocapsid protein-coding mRNAs was observed following 24 h of CBD treatment. It also suggested that CBD acts to prevent viral protein translation and the associated cellular changes [98]. As underlying mechanisms, CBD may suppress viral infection and promote viral RNA degradation through the induction of the interferon signaling pathway. CBD may also reduce the viral titer to allow the normal host activation of the interferon pathway, which is suppressed upon infection by SARS-CoV2 [99,100].

CBD is metabolized to 7-carboxy-cannabidiol (7-COOH-CBD) and 7-hydroxy-cannabidiol (7-OH-CBD) in the liver and intestine. CBD and 7-OH-CBD inhibit SARS-CoV2 replication in A549-ACE2 cells (EC50 of 3.6 μM) [98]. The levels of these cannabinoids in healthy patients taking CBD (Epidiolex ^®^) show a maximum concentration (C_max_) of CBD in blood plasma in the nM range, while 7-OH-CBD has a Cmax at μM concentrations. These results suggest that CBD itself is not present at sufficient concentrations for the effective inhibition of the SARS-CoV2 transmission cycle in humans. In contrast, the plasma concentrations of its metabolite 7-OH-CBD, whose C_max_ is augmented following co-administration of CBD with a high-fat meal, are sufficient for the potential blockade of human SARS-CoV2 infection [101].

The crystal structure of the SARS-CoV2 spike protein (C-terminal domain) in complex with the ACE2 receptor (human) was solved in 2020, giving the opportunity to understand the interaction between the virus and the human receptor [102]. This knowledge gives opportunities to create selective ligands targeting the viral S1 protein and preventing human cells from SARS-CoV2 infection [5]. In this sense, cannabidiolic acid (CBDA) and cannabigerolic acid (CBGA) showed high binding affinity to recombinant S1 [103].

On the other hand, in silico analysis suggests that CBD binds to the S1 virus membrane protein by interactions with the residues Q189, M165 and E16, causing an inhibitory effect with an IC_50_ of 7.91 µM [104]. On the other hand, ACE2 expression in the epithelium is high in males, older people and smokers, evidencing the risk of the population to the expression of the ACE2 receptor [105]. Downregulation of ACE2 levels due to CBD actions could be a strategy to decrease susceptibility for virus infection. CBD decreases the expression of ACE2 and TMPRSS2 in oral, lung and intestinal epithelia, which could reduce the burden of the virus [71,106,107]. A study conducted in different tissues (EpiAirway TM, EpiAirwayFT TM, EpiOral TM, EpiIntestinal TM) with a level of ACE2 and TMPRSS2 overexpression identified the significant effects of cannabis in decreasing ACE2 and TMPRSS2 expression levels [107]. This information suggests that phytocannabinoids may prevent SARS-CoV2 infection.

To determine the mechanisms by which the expression downregulation of ACE2 and TMPRSS2 takes place, the mRNA levels of the genes miR-200c-3p and let-7a-5p encoding ACE2 and TMPRSS2 were measured. qRT-PCR showed that CBD modulated ACE2 and TMPRSS2 expression levels through transcriptional and post-transcriptional mechanisms. A possible mechanism by which they obtain these effects is through the activation of the AKT pathway participating in the post-transcriptional regulation of ACE2 and TMPRSS2 expression. CBD significatively decreased p65 (p-p65, Ser536) and phosphorylated AKT1/2/3 (p-AKT1/2/3) levels [108]. 

On the other hand, CBD is capable of significantly inhibiting the human hepatic microsomal metabolism of Remdesivir (RDV), the first drug for viral therapy to treat SARS-CoV2/MERS-CoV infection, extending the half-life of RVD through the remarkable inhibitory activity of human liver carboxylesterase-1 (CES1) and CYP3A4 microsomal metabolism. These findings suggest that CBD could be an adjuvant to RDV in the treatment of SARS-CoV2 infection [109].

## 8. Cytokine Storm and the Anti-Inflammatory Activity of CBD

Viral proteins and RNA from the virus genome accumulate, forming virons in the endoplasmic reticulum and the Golgi apparatus and are then transported in vesicles to the extracellular space [104]. In this process, proinflammatory M1 macrophages and helper T cells secrete interleukins, which produce inflammation within lung cells. Airway epithelial cells are known to respond to the effects of cannabinoids, which are dependent or independent of CB2 receptor activation [110]. In this inflammatory stage, CB2 receptors activated by CBD inhibited inflammatory processes such as macrophage migration to the lungs. This may be due to cannabinoid CB2 receptor overexpression in immune cells, e.g., CD8 lymphocytes, CD4 lymphocytes, monocytes and neutrophils in the spleen, thymus, kidney, lung, liver, nasal epithelium, and brain [111]. Furthermore, the stimulation of this receptor decreased the release of proinflammatory cytokines IL-1, IL-6, IL-8, IL-12 and TNF-α, by suppressing TNFα, IFNγ and IL-6 and IL-8 production as well as the transcription of COX-2 [4,69,108,112]. Moreover, CB2 agonists augmented IL-10 expression with anti-inflammatory characteristics [113]. Subsequently, CBD is capable of regulating lung cell inflammation and respiratory stress [114].

Polymorphonuclear leukocytes (PMN) produce and release cytokines and chemokines as the first line of defense against pathogen microorganisms; however, they often cause tissue damage. Cosentino et al. demonstrated that an extract standardized to 5% CBD and low THC content (<0.2%) effectively inhibited cell migration and oxidative metabolism in human PMN [115]. Another in vitro study demonstrated the effect of the extracts of inflorescences of the Arbel variety of *C. sativa* (with high CBD content) on the pulmonary epithelial inflammation that develops in the severe phase of the disease. The extract has significant activity in reducing cytokine secretion levels that characterize the cytokine storm (IL-6 and IL-8) in pulmonary epithelial cells, with IC_50_ values of 3.45 µg/mL and 3.49 µg/mL, respectively. In an experimental asthma rat model, CBD diminished airway inflammation and IL-4, IL-5, IL-13, IL-6, IL-7 and TNF-α serum levels, which contribute to fibrosis and asthma pathophysiologies [116,117]. In addition, CBD reduced the secretion of IL-1 and IFNγ by T cells [118]. Kovalchuk et al. analyzed several cannabinoid extracts, showing that six of them diminished Toll-like receptor 2 (TLR2), NFKB2, CCL2 (known as MCP-1) and WNT-2 and WNT-5a expression levels. All these proteins are involved in lung disease, ARDS, expression of further proinflammatory genes and fibrosis [115].

## 9. Cannabis and COVID-19: In Vivo Studies

Currently, apart from supportive measures for COVID-19 patients, there is no definitive cure for ARDS. This demonstrates the urgent necessity for effective therapeutics to treat this complex condition. An interesting in vivo study was performed by Khoadadadj et al., in which they reproduced the histopathological and viral features of ARDS associated with SARS-CoV2 infection using a synthetic high molecular weight analog of double-stranded RNA, Poly(I:C) polycytidylic acid Poly(I:C). CBD treatment resulted in the reduced presence of lymphocytes, neutrophils and monocytes and the levels of proinflammatory cytokines (e.g., IL-6, IFNγ and TNFα) [119]. In parallel, by using intranasal administration of the synthetic analog Poly (I:C) in a mouse model, another research group demonstrated that CBD can also improve ARDS symptoms through a significant increase in both blood and lung tissue apelin expression, a peptide with an important role in the central and peripheral regulation of immunity and the central nervous, metabolic and cardiovascular systems [120].

Nuclear PPARγ receptors are overexpressed in alveolar macrophages and responsible for cytokine hypersecretion resulting in the amelioration of tissue damage. CBD and other cannabinoids modulate PPARγ receptor activity, reducing pulmonary inflammation and promoting lung recovery after viral infections [121]. CBD, as a PPARγ receptor agonist, may show direct antiviral activity by regulating fibroblast/myofibroblast activation, potentially limiting the onset of late-onset pulmonary fibrosis in cured COVID-19 patients [122]. In addition, the intraperitoneal administration of CBD improved lung function and reduced inflammation in experimental acute lung injury (ALI) [123,124], pulmonary hypertension [125], hypoxic/ischemic brain injury-induced lung injury [126] and asthma [116,117]. A clinical trial demonstrated that 300 mg/day of oral-administered CBD diminished or prevented the deterioration from mild/moderate to severe/critical clinical status as determined by the COVID-19 scale or the natural course of evolution of typical clinical symptoms [127]. In humans, the use of cannabinoids prevented the induction of proinflammatory CD16 monocyte and IP-10 production, showing anti-inflammatory effects [55]. Moreover, cannabis use decreased CD4 and CD8 T cell numbers as well as the inflammatory IL-10, IL-12 and TNF-α production compared to non-cannabis users [72]. An overview of CBD functions in SARS-CoV2 infection is presented in Figure 5. 

## 10. Cannabidiol and the Psychological Effects of COVID

The COVID-19 pandemic resulted in devastating mortality and morbidity. Billions of lives were altered in a socioeconomic and health scale, including respiratory, cardiovascular and mental health consequences such as anxiety, depression and substance use after recovery [128]. In the U.S., 45% of affected adults showed mental health problems because of stress from the virus, job loss or imposed shelter-in-place [129]. Phytocannabinoids such as CBD could be an alternative therapy for panic disorders, anxiety, post-traumatic disorder (PTSD) and/or depressive disorders [51,130].

Within the endocannabinoid system, the highest density of CB1 receptors is distributed in key areas for stress and emotion regulation such as the prefrontal cortex (PFC), the hippocampus and the amygdala [131]. Since the development of the COVID-19 pandemics, many people have presented high levels of stress, fear and anxiety as the result of an anormal CB1 receptor expression in the amygdala, nucleus accumbens (reward system) and in the PFC, the latter being associated with increased levels of aggressive behavior [132,133]. Chronic stress conditions are characterized by low levels of the endocannabinoid anandamide; such effects are apparently more pronounced in women than in men [134]. During the global COVID-19 pandemic, women have shown higher rates of anxiety and depression than men, due to an increased domestic overload, domestic violence and impoverishment [135,136].

CBD reduced euphoria and depressive and psychotic symptoms [137] as well as reducing anxiety and stress-induced reactions in healthy volunteers [138,139,140]. It showed beneficial effects in patients with naïve social anxiety disorder [141,142], post-traumatic stress disorder [143], psychiatric symptoms [144] or at high risk for psychosis [145,146]. Therefore, the anxiolytic and antinociceptive properties of CBD make this compound an important adjuvant treatment for improving the well-being and life quality of COVID-19 patients. Cannabinol may even be applied after recovery from COVID-19 infection to alleviate post-traumatic stress symptoms.

In a randomized clinical trial, 300 mg of CBD (150 mg twice a day) was administered to healthcare professionals caring for COVID-19 patients. The results showed that CBD reduced burnout and emotional exhaustion symptoms [147]. Another study revealed that CBD may also benefit wakefulness by increasing the levels of dopamine in the lateral hypothalamus or dorsal raphe nuclei, which are responsible for wakefulness, suggesting that CBD may be used for sleep disorder therapy [148], as one of the symptoms that appear after contracting COVID. As discussed in this work, cannabinoids, especially CBD, can efficiently ameliorate or attenuate this type of pathology.

## 11. Conclusions

Cannabinoids, especially CBD, appear to be promising in the treatment of COVID-19, as an adjuvant of current antiviral drugs, reducing lung inflammation by decreasing chemokines and cytokines secreted by the cells of the immune system or mediating in the CNS reducing morbidity as fear, anxiety, stress, sleep disorders. However, more research and clinical studies are necessary, especially to establish the effects of their long-term use. In any case, many countries are allowing the use of medical cannabis and this plant, which has been used since ancient times, could be a natural therapeutic alternative for COVID-19 infected patients, but there is still a long way to go for its acceptance and use in routine clinical practice.

## Figures and Tables

**Figure 1 life-12-02117-f001:**
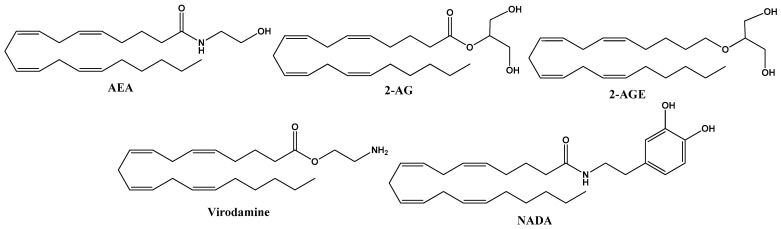
Chemical structure of endocannabinoid derivates from arachidonic acid, including anandamide (AEA), 2–arachidonoyl glycerol (2–AG), 2-arachidonoyl–glycerol (2–AGE), O-arachidonoyl-ethanolamine (virodamine) and N-arachidonoyl-dopamine (NADA).

**Figure 2 life-12-02117-f002:**
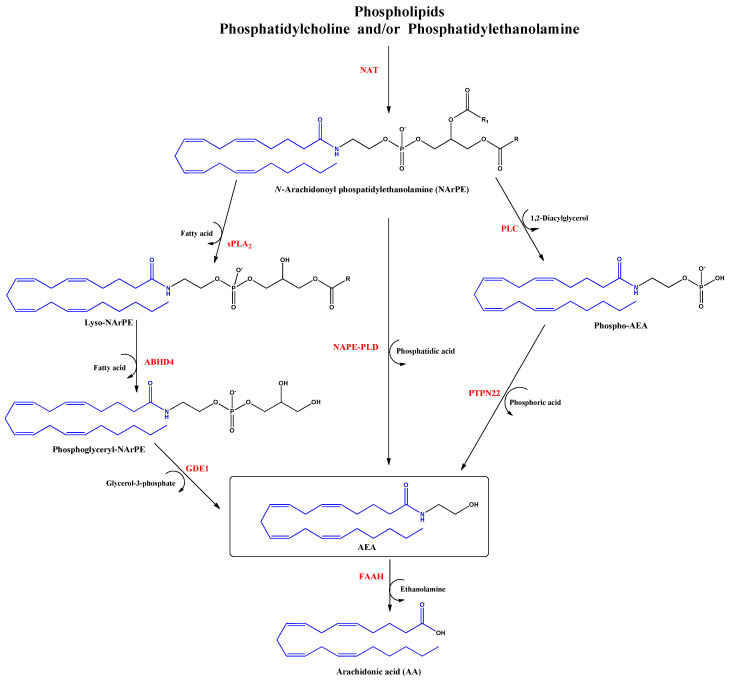
Biosynthesis and hydrolysis of N–Arachidonoylethanolamine (AEA) pathway. Abbreviations in red mean the enzymes: NAT: N–acyltransferase; sPLA2: soluble phospholipase A2; PLC: phospholipase C; ABHD4: α/β–hydrolase domain 4; GDE1: glycerophosphodiesterase; NAPE–PLD: NAPE–specific phospholipase D; PTPN22: non–receptor protein tyrosine phosphatase 22; FAAH: fatty acid amide hydrolase.

**Figure 3 life-12-02117-f003:**
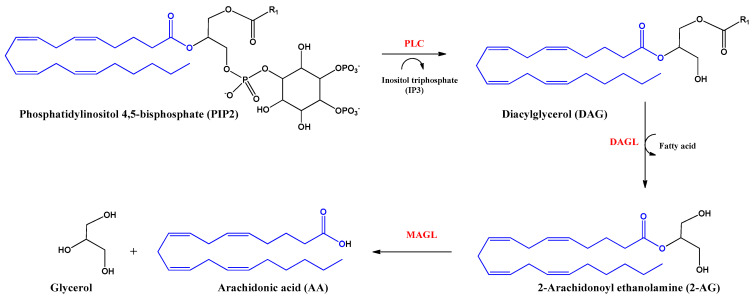
Biosynthesis and hydrolysis of 2–Arachidonoyl ethanolamine (2–AG) and arachidonic acid (AA). Abbreviations in red mean the enzymes: PLC: Phospholipase C–β; DAGL: Diacylglycerol lipase; MAGL: monoacylglycerol lipase.

**Figure 4 life-12-02117-f004:**
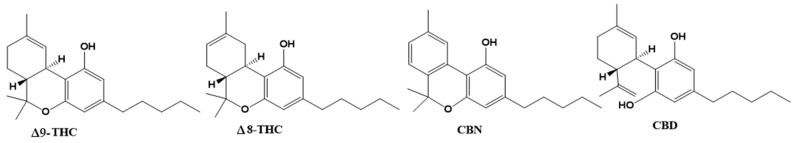
Chemical structures of the phytocannabinoids Δ9–tetrahydrocannabinol (Δ9-THC), Δ8–THC, cannabinol (CBN) and cannabidiol (CBD).

**Figure 5 life-12-02117-f005:**
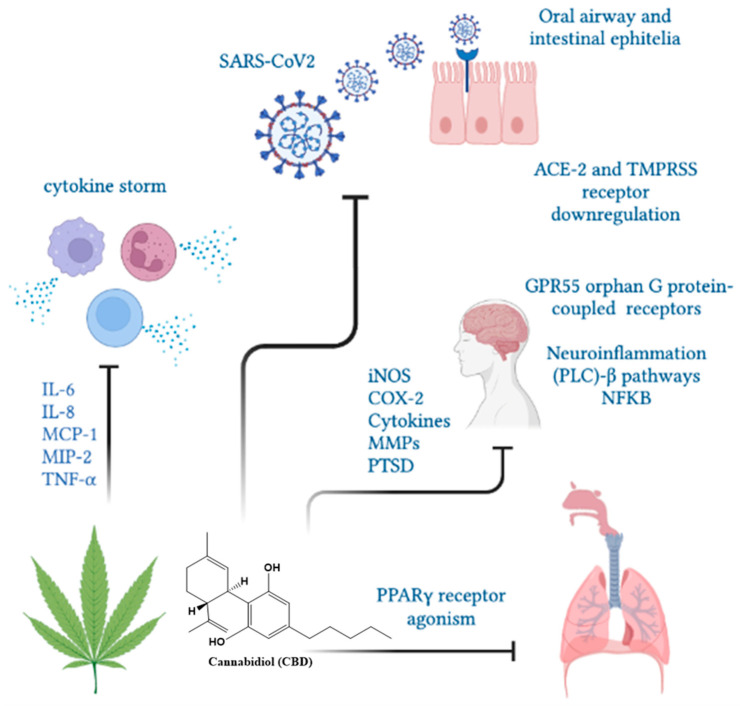
The CBD functions in SARS-CoV2 infection. Cannabinol could limit the severity of SARS-CoV2 infection by decreasing ACE-2 receptors, cytokine mitigation and lung inflammation and fibrosis through PPARγ receptors.

**Table 1 life-12-02117-t001:** Commercial formulations of phytocannabinoids with therapeutic properties.

Commercial Name	Active Ingredient	Therapeutic Applications	Countries, in Which Its Use Is Approved
MARINOL^®^ [44,45,46]	Capsules of 2.5 mg, 5 mg or 10 mg of dronabinol (THC) dissolved in sesame oil.	Treatment of nausea and vomiting due to chemotherapy, in adults and pediatric patients. Treatment of anorexia/cachexia in adults with AIDS or terminal cancer.	United States, Canada, South Africa, Denmark.
SATIVEX^®^ [45,46,47]	Spray of a standardized extract of Cannabis. Each spray releases a fixed dose of 2.7 mg of THC and 2.5 mg of CBD.	Adjuvant treatment for: 1. moderate or severe spasticity due to Multiple Sclerosis (MS); 2. symptomatic pain relief neuropathic in adult patients with MS; 3. in adult cancer patients suffering from moderate to severe pain.	Approved in 27 countries (European Nations and Asia, New Zealand, Canada and Australia).
Epidiolex^®^ [45,46,48]	Oral solution (100 mL of 100 mg/mL CBD	Reduces seizure frequency in patients with end-stage Dravet syndrome, a rare and devastating form of epilepsy that is drug-resistant.	GW Pharmaceuticals (United Kingdom)
Nabiximol [49]	Mouth spray (50% CBD and 50% THC)	Treatment of spasticity in multiple sclerosis.	U.S., Canadian and U.K. regulatory agencies

**Table 2 life-12-02117-t002:** Pharmacological efficacy of cannabinoid products tested in clinical trials.

Clinical Trial Number/Year	Patients	Description	Results	References
NCT02224560/2018	Adults and children (2–55 years of age, *n* = 225) with Lennox-Gastaut syndrome	Randomized, double-blind trial 14-week study of two dose levels (10 mg/kg/day and 20 mg/kg/day), to evaluate the efficacy and safety of CBD (GWP42003-P)	It confirmed a reduction in the number of seizures	[59]
NCT02224690/2015	Patients (2-55 years of age) with Lennox-Gastaut syndrome (*n* = 171)	GWP42003-P in a single dose of 20 mg/kg/day for 14 weeks	The results confirmed the reduction in seizure frequency after administration	[60]
NCT02091206/	Children (4–10 years) suffering from Dravet syndrome (*n* = 34)	It was a safety and pharmacokinetic dose-ranging study of Epidiolex (5, 10 and 20 mg/kg/day) with a duration of 21 days	Results confirmed that Epidiolex is a safe drug; however, it caused more adverse effects than placebo did	[61]
NCT02091375/ 2015	Children and adults (2–18 years) with Dravet syndrome (*n* = 120)	Evaluation of the antiepileptic efficacy of GWP42003-P	Results confirmed that CBD (20 mg/kg/day/14 weeks) resulted in 43% of the patients experiencing a reduction in seizure frequency of 50% or more. Three of these patients became seizure-free	[62]
NCT02700412/ 2015	Eighty epileptic subjects between 18 and 99 years	This interventional study evaluated the safety and tolerability of Epidiolex at various doses between 5 and 50 mg/kg/day as an add-on drug for the treatment of drug-resistant debilitating epilepsy	Results confirmed Epidiolex seizure reduction in 63.6% of patients after 12 weeks of treatment	[63]
NCT02695537/ 2015	Children and adults (1–19 years), *n* = 89	Evaluation of the safety and tolerability of Epidiolex for the treatment of drug-resistant debilitating epilepsy	The results confirmed after administration of various doses between 5 and 50 mg/kg/day a seizure reduction of 63.6% after 12 weeks of treatment	[63]

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
