# Peer review of "Therapeutic Effects of Cannabinoids and Their Applications in COVID-19 Treatment"

_life, 2022, doi:10.3390/life12122117_

Round 1

Reviewer 1 Report

Please, refer to the attached file. Thank you.

Author Response

Dear reviewer, we would like to thank your valuable comments that increase the quality  of this review.  

All your reviewer's suggestions were considered and relevant changes were performed. While not requested, we provide a new graphical abstract. We hope that the manuscript meets now the quality of publication in Life

Reviewer 1 Report

Authors highlight the potential of cannabinoids derived from Cannabis sativa for the treatment of Covid-19. In view of the urgency in the scientific world to find effective remedy for the pandemic, it is refreshing as this review opens up some therapeutic avenues for Covid-19. Notwithstanding, authors need to improve the following:

  1. Language improvements, e.g., Recently, however, suppressive power; the Figure 1.; known of the pharmacological plant activity; In spite of that THC displays

Response: English writing style has been improved, as suggested by the reviewer

  1. Define first all abbreviations, then use subsequently, e.g., THC and CBD

Response: The description of all compounds was done in the introduction because we don´t have enough space in the abstract: line 41; 44; 80.

  1. Introduction needs improvement, e.g., the first sentence appears incomplete. Lack of standard treatment for what condition?

Response: We add a new sentence in this part

  1. Check this - arachidonic acid as endogenous cannabinoids present in the organism of all animals, including humans.

Response: we deleted the sentence including humans.

  1. Check spelling – phosphatyletanolamine

Response: checked

  1. Chemically is the esterification product of arachidonic acid and ethanolamine. – what is the

esterification product of arachidonic acid?

Response: we deleted that sentence

  1. Refine - It behaves as a full agonist or partial CB1 receptor agonist as well as a partial CB2.

Response: corrected

  1. Countries such as China, India, Africa, Turkestan and Arabia – Africa is not a country

Response: we deleted Africa

  1. Factual statements should have a reference - Countries such as China, India, Africa, Turkestan and Arabia have used its compounds for medicinal purposes to treat diseases such as malaria, beriberi, constipation, rheumatic pains, headaches, female ailments and other conditions.

Response: We added a reference

  1. Check this - flowers of the female plant. (31)

Response: Female was delated

  1. Use a table to summarize the clinical studies or evidence of the use of cannabinoids

Response: We provide an additional table with this information.

  1. Check this - administered. (66).

Response: checked and corrected to “applied”

  1. Improve - The main receptor of SARS-Cov2

Response: Henning please check this sentence that I added in the line 258 The angiotensin-converting enzyme 2 (ACE 2) has been pointed out as the principal receptor in the interaction of SARS-Cov 2 with human cells

  1. Improve - being these the most infected

Response: We change the sentence to being these cells more susceptible to be infected by the virus

  1. Check - inflammation are overexpressed increased expression of chemokines

Response: We change the sentence, line 273

  1. Some sentences are too long, e.g., line 301…

Response: We have shortened the sentences as requested.

  1. Revise - anti-inflammatory power of CBD

Response: we change the title to: Cytokine storm and  anti-inflammatory activity of CBD

  1. Check - CB2 receptor-independent. (111). Line 377

Response: corrected to “which are dependent or independent of CB2 receptor activation”

  1. Check the spelling of cytokine in Figure 5. It is not “cytoquine”

Response: Corrected to Cytokine

  1. Check and improve - This work has been supported

Response: checked

  1. Properly format in-text citations

Response: Done

  1. Reconcile in-text citations with reference list

Response: We checked the references

Reviewer 2 Report

The concise title of the review article submitted for publication by Rebeka PEREZ and her colleagues accurately reflects the topic of the paper: ’Cannabinoids for Covid 19’  treatment. Unfortunately, the Covid epidemic has been part of our lives for the third year now, so the actuality and importance of the topic cannot be disputed. On the other hand, we must also see that the judgement of cannabinoid active substances has changed significantly in recent years. Cannabinoids, like opioids, are active substances with a Janus face, since they can also serve as recreational and plesaurable drugs, medicines, and drugs of abuse. The medical and social perception of THC remains controversial and can vary greatly from country to country. However, the beneficial effects of the other important phytocannabinoid cannabidiol (CBD) are becoming increasingly apparent. The paper  provides a well-articulated and thorough overview of endo- and phytocannabinoids, cannabinoid receptors, and the in vitro and in vivo effects of phytocannabinoids in the case of Covid virus infection. The carefully edited color figures complement the textual information of the article well. I definitely support the acceptance of the article, but I have some suggestions for the final version:

The word ’Cannabis’ is the name of the hemp plant, moreover the collective name for the active substances that can be extracted from the plant. It would be worth mentioning the two most common forms of cannabis drugs, hashish and marijuana.

Line 165:  Cannabis sativa is also a common plant in temperate climate zones, e.g. in Europe.

Line 175:  ’The Cannabis family of plants contains over 500 different compounds…’  Cannabis is genus, Cannabinaceae or  Cannabaceae the ’family’ name in the botany. I suggest:  ’over 500 bioactive compounds’, since the plants certainly contain many more compounds than this.

Line 196:    CDB is wrong, please correct it as CBD .    CDB is mentioned in two other places in the text, please find it and change it  J .   The other 75 CBD are OK J

Figure 5:  The compound and protein names are colored blue or red, I think their meanings are different, this should be explained either in text or in the figure legend.

Author Response

Dear reviewer, we would like to thank your valuable comments that increase the quality  of this review.  

All your suggestions were considered and relevant changes were performed. While not requested, we provide a new graphical abstract. We hope that the manuscript meets now the quality of publication in Life.

Reviewer2 Report

The concise title of the review article submitted for publication by Rebeka PEREZ and her colleagues accurately reflects the topic of the paper: ’Cannabinoids for Covid 19’  treatment. Unfortunately, the Covid epidemic has been part of our lives for the third year now, so the actuality and importance of the topic cannot be disputed. On the other hand, we must also see that the judgement of cannabinoid active substances has changed significantly in recent years. Cannabinoids, like opioids, are active substances with a Janus face, since they can also serve as recreational and plesaurable drugs, medicines, and drugs of abuse. The medical and social perception of THC remains controversial and can vary greatly from country to country. However, the beneficial effects of the other important phytocannabinoid cannabidiol (CBD) are becoming increasingly apparent. The paper  provides a well-articulated and thorough overview of endo- and phytocannabinoids, cannabinoid receptors, and the in vitro and in vivo effects of phytocannabinoids in the case of Covid virus infection. The carefully edited color figures complement the textual information of the article well. I definitely support the acceptance of the article, but I have some suggestions for the final version:

  • The word ’Cannabis’is the name of the hemp plant, moreover the collective name for the active substances that can be extracted from the plant. It would be worth mentioning the two most common forms of cannabis drugs, hashish and marijuana.

Response: We added a sentence in the lines 31-32

  • Line 165:  Cannabis sativais also a common plant in temperate climate zones, e.g. in Europe.

Response: we revised the text according to  your suggestion

  • Line 175:  ’The Cannabis family of plants contains over 500 different compounds…’  Cannabisis genus, Cannabinaceae or  Cannabaceae the ’family’ name in the botany. I suggest:  ’over 500 bioactive compounds’, since the plants certainly contain many more compounds than this.

Response: we changed the sentence following your suggestions

  • Line 196:    CDB is wrong, please correct it as CBD .    CDB is mentioned in two other places in the text, please find it and change it  J .   The other 75 CBD are OK J

Response: Thank you. We made the correction.  

  • Figure 5:  The compound and protein names are colored blue or red, I think their meanings are different, this should be explained either in text or in the figure legend.

Response: The figure was changed.

Reviewer 3 Report

An excellent review, but there should be a greater focus on Covid 19, like the title

Author Response

Dear reviewers, we would like to thank your valuable comments that increase the quality  of this review.  

 All your reviewers’ suggestions were considered and relevant changes were performed. While not requested, we provide a new graphical abstract. We hope that the manuscript meets now the quality of publication in Life.

Reviewer 3 Report:

  • An excellent review, but there should be a greater focus on Covid 19, like the title

Response: Thank you for your comment. We would like to maintain the present version of the manuscript, as we introduced therapeutic effects of cannabinoids and then focused on their applications in  Covid-19  treatment.